# Dermoscopic Image Classification Method Using an Ensemble of Fine-Tuned Convolutional Neural Networks

**DOI:** 10.3390/s22114147

**Published:** 2022-05-30

**Authors:** Xin Shen, Lisheng Wei, Shaoyu Tang

**Affiliations:** 1School of Electrical Engineering, Anhui Polytechnic University, Wuhu 241000, China; k168sx@163.com (X.S.); tangshy2022@163.com (S.T.); 2Anhui Key Laboratory of Electric Drive and Control, Anhui Polytechnic University, Wuhu 241002, China

**Keywords:** dermoscopic image, classification, deep learning, transfer learning, fine-tuning, ensemble learning

## Abstract

Aiming at the problems of large intra-class differences, small inter-class differences, low contrast, and small and unbalanced datasets in dermoscopic images, this paper proposes a dermoscopic image classification method based on an ensemble of fine-tuned convolutional neural networks. By reconstructing the fully connected layers of the three pretrained models of Xception, ResNet50, and Vgg-16 and then performing transfer learning and fine-tuning the three pretrained models with the ISIC 2016 Challenge official skin dataset, we integrated the outputs of the three base models using a weighted fusion ensemble strategy in order to obtain a final prediction result able to distinguish whether a dermoscopic image indicates malignancy. The experimental results show that the accuracy of the ensemble model is 86.91%, the precision is 85.67%, the recall is 84.03%, and the F1-score is 84.84%, with these four evaluation metrics being better than those of the three basic models and better than some classical methods, proving the effectiveness and feasibility of the proposed method.

## 1. Introduction

Melanoma is an extremely dangerous form of skin cancer. The United States reports more than 10,000 new cases a year, including more than 9000 deaths [1]. In Europe, melanoma kills more than 20,000 people every year [2]. However, if detected early and diagnosed in time, the cure rate is very high [3].

Dermoscopy is a non-invasive imaging technique that can obtain an enlarged, well-lit image of a local area of the skin while eliminating reflections on the skin surface and enhancing the image clarity of the skin lesions. Compared with traditional visual diagnosis, this dermoscopic diagnostic method can greatly improve accuracy. Although dermoscopy can better visualize skin lesions and improve sensitivity and specificity compared to visual inspection, dermatologists have struggled to achieve higher diagnostic performance with skin lesions. This is because the manual examination by experts of dermoscopic images is often tedious, error-prone, complex, subjective, and time-consuming [4]. As a result, automated computer-aided diagnosis (CAD) systems have become an essential diagnostic tool to support and assist experts in clinical decision-making. In the whole detection process of the dermoscopic images, the last and extremely critical step is the accurate classification of the dermoscopic images. Due to the size, texture, and shape of the lesion area, and the existence of various artifacts, the classification of dermoscopic images brings challenges. 

In the field of dermoscopic image classification, scholars have proposed many related detection algorithms, such as support vector machines and other traditional machine learning methods, the effect of which needs to be further improved [5]. If a dataset for an intelligent auxiliary diagnosis system for dermoscopic images can be developed, diagnosis can be made and the medical treatment strategy can be determined as long as the image of the skin lesion area is provided for diagnosis, which has very important theoretical research and practical value. As an important branch of artificial intelligence, medical image processing uses deep-learning algorithms to automatically detect the lesion area in a dermoscopic image, improve the accuracy of lesion detection, reduce the misdiagnosis rate, and improve the detection efficiency in dermoscopic imaging, which has become a hot research direction for experts and scholars. However, there are still many problems that need to be solved. On the one hand, the effect of using convolutional neural networks for dermoscopic image classification depends on the performance of the network, and it is still a difficult problem to design or select an appropriate network model for a specific task. However, the current algorithms often use a single convolutional neural network, so the classification performance of such models is low, with very few exceptions. On the other hand, deep convolutional neural networks need to be trained with a large number of data samples to complete the accurate classification of objects. In the field of medical imaging, it is very difficult to collect large samples. An insufficient number of learning samples results in the features learned by the deep learning model not being robust and representative and also results in poor generalization ability for the model. For the problem of the difficulty of model training due to the small number of datasets, the use of transfer learning is a novel and effective method. Therefore, in view of the above problems and the current research status, this paper proposes a dermoscopic image classification algorithm integrating fine-tuning convolutional neural networks by using the three models of Xception, ResNet50, and VGG16 pretrained on the “ImageNet” dataset as the basic model of the integrated model, and then fine-tuning the models on the ISIC 2016 Challenge official skin dataset by reconstructing the fully connected layers of the three pretrained models. Finally, the final ensemble model is obtained by the weighted fusion method of the model output class probability. The related experimental results show that the overall performance of the ensemble model is better than the three base models and some other traditional methods.

The main structure of this paper is as follows: the first part is the introduction, which briefly introduces the research background, development status, and research significance of this paper. The second part is the related work, which introduces some of the recent efforts in the field. The third part introduces the research method of the proposed algorithm, and the fourth part is the experimental simulation of the research method and the explanation of the experimental results. Finally, the fifth part discusses the experimental results and the sixth part gives a brief conclusion.

## 2. Related Work

The traditional detection of skin diseases is initially judged by the human eye, mainly including the pattern analysis method proposed by Pehamberger H et al. [6], the seven-point detection method proposed by Argenziano G [7], and the ABCD rule proposed by Stolz W. et al. [8]. These methods basically rely on the manual analysis of specific features to assist in diagnosis. Although they can achieve a good diagnostic effect, they require a lot of energy and have certain limitations. With the development of computer technology, computer-aided diagnosis has been introduced to the field of dermoscopic image detection. The computer-aided diagnosis system can screen a large number of dermoscopic images and give diagnostic opinions within the allowable error range, which greatly reduces the workload of dermatologists and reduces unnecessary pathological analysis. Computer-aided diagnosis of melanoma initially employed traditional machine learning methods. For example, in image preprocessing, a hair removal algorithm, proposed by Lee T et al. [9] and then Bi L et al. [10], was used to remove hair from the dermoscopic images in their own work. In feature extraction, Celebi ME et al. [11] extracted relevant features such as shape, color, and texture from images and then used a variety of feature selection algorithms to sort the features and input the top-ranked features into the support vector machine (SVM) for classification. In the classification and recognition stage, Ballerini L et al. [12] used the K-nearest neighbor model for classification and recognition after extracting color and texture information. The work of the above scholars is very instructive, but they only rely on low-dimensional information for classification, such as the color and texture of the images, and this type of method can only further improve the classification and recognition ability of these diagnostic systems to a certain extent.

In recent years, with the rapid development of deep learning, major breakthroughs beyond human capabilities have been achieved in many fields, such as image classification and speech recognition. In the classification task, the first proposed convolutional neural network was LeNet [13], and then many famous convolutional neural networks were proposed successively, such as GoogleNet [14], ResNet [15], DenseNet [16], etc., and these networks perform well in classification tasks. In the target detection task, the main representatives are Fast-RCNN [17], proposed by Girshck R et al., and Faster-RCNN [18], proposed by Ren S et al. These methods improve the accuracy of target detection. In the field of dermoscopic image detection, Kawahara et al. [19] used a CNN network to identify skin cancer images and used the Dermofit skin cancer dataset to train, with am obtained diagnostic accuracy of 78.1%. Pomponiu et al. [20] proposed using a pretrained deep neural network to automatically extract a set of representative features and then use the k-nearest neighbor classifier (KNN) to classify them. Esteva A et al. [21] published an article on the use of deep-learning methods for skin cancer disease classification in Nature in 2017, and they integrated multiple skin disease image datasets, covering almost all skin cancer types, and directly used the GoogleNet pretraining model for transfer learning, with the accuracy of the results obtained being slightly higher than the diagnostic accuracy of dermatologists, thus verifying the feasibility of deep neural networks in skin cancer classification tasks. Gal et al. [22] proposed a combination of Bayesian learning and active learning frameworks to identify skin cance. Ashraf et al. [23] proposed a region-of-interest (ROI)-based system to detect melanoma using an off-the-shelf deep CNN model which extracted ROI from dermoscopic images by utilizing a modified k-means algorithm, and then used the AlexNet model and enhanced ROI images to accurately identify melanoma skin cancer. Daghrir et al. [24] proposed a hybrid approach to detect melanoma by using a convolutional neural network and two machine learning classifiers. These models were trained using various types of features, such as the color, texture, and shape of skin lesions. Finally, an ensemble method based on majority voting was adopted to improve the performance. Mahbod et al. [25] proposed and evaluated a multiscale multi-CNN (MSM-CNN) fusion approach based on a three-level ensemble strategy that utilized the three network architectures trained on cropped dermoscopic images of various scales. Zhang et al. [26] proposed an attention residual-learning convolutional neural network (ARL-CNN) model for skin lesion classification in dermoscopic images, which is composed of multiple ARL blocks, a global average pooling layer, and a classification layer. Each ARL block jointly uses the residual learning and a novel attention learning mechanism to improve its ability in discriminative representation. Srinivasu et al. [27] proposed the MobileNet V2 with an LSTM component for the purpose of the precise classification of skin disease from images captured from mobile devices. The practical implication of the model is to design the app through which the image of the affected region of the skin is captured in order to determine the kind of skin disease shown in the image. Kousis et al. [28] proposed 11 CNN (convolutional neural network) candidate single architectures. They trained and tested those 11 CNN architectures by using the HAM10000 dataset, comprised of seven skin lesion classes. From the 11 CNN architecture configurations, DenseNet169 produced the best results. It achieved an accuracy of 92.25%, a recall (sensitivity) of 93.59%, and an F1-score of 93.27%, which outperforms existing state-of-the-art efforts. Anand et al. [29] proposed a transfer learning-based model with the help of the pretrained Xception model. The Xception model was modified by adding layers such as one pooling layer, two dense layers and one dropout layer. A new fully connected (FC) layer expanded the original fully connected (FC) layer based on seven skin disease classes. The proposed model was evaluated on a HAM10000 dataset with large class imbalances. Data augmentation techniques were applied to overcome the unbalancing in the dataset, and the new results show that the model attained an accuracy of 96.40% for classifying skin diseases.

## 3. Materials and Methods

The algorithm flowchart of the proposed method is shown in Figure 1. 

In Figure 1, firstly, the dermoscopic image is preprocessed, including adjusting the size and color correction of the dataset image. Secondly, data enhancement is performed, including the horizontal and vertical flipping of the dermoscopic image; the Xception, ResNet50, and Vgg-16 models are fine-tuned on the dataset, and then the output results are integrated by the integration strategy of weighted fusion. Finally, the performance of the integrated model is tested and evaluated on the test set, and the final classification result is generated.

### 3.1. Dataset

The dataset used in this method is the skin disease image dataset publicly used by the International Skin Imaging Collaboration Organization 2016 Challenge. Compared with other updated datasets, the ISIC 2016 Challenge official skin dataset has a small amount of data and is unbalanced, which is more challenging for model training. The ISIC 2016 Challenge official skin dataset includes 900 training-set pictures and 379 test-set pictures. During the experiment, this paper randomly selects 20% of the 900 training images as the validation set, adjusts the model with reference to the validation results, and uses the remaining 80% as the training-set data. The distribution of the two types of dermoscopic images in the experimental dataset is shown in Table 1.

### 3.2. Image Preprocessing

First, the resolution of the dataset images is adjusted. Due to the inconsistent resolution of the pictures in the ISIC 2016 Challenge dataset, it has a great impact on the training of the model, and the convolutional neural network used in this paper needs a resolution of 224 × 224. So in order to fit the network model, prevent the gradient explosion problem in the training process, and keep the input of the network consistent, the resolution of the dataset pictures is uniformly adjusted to 224 × 224 pixels.

Secondly, since the ISIC 2016 Challenge dataset was not collected under standard conditions, it may contain data from different institutions; the shooting conditions are also very different, so the situation of large color differences in the picture often occurs, and this kind of image is called a multi-source image. In order to solve the image-color deviation caused by the acquisition environment and equipment, some color correction methods can be used. Even under different lighting conditions, the human visual system can still distinguish the true color of things, that is, their color constancy, which is widely used in dermoscopic image recognition. In this paper, the Shades-of-Gray algorithm is used to correct the color constancy of the image. Comparisons between selected benign melanoma images and malignant melanoma images processed by the above image preprocessing method and the original image are shown in Figure 2.

It can be seen from the comparison in Figure 2 that after the preprocessing of the proposed shadows of the Shades-of-Gray algorithm, the lesion area of the image is more obvious and the color between the images is more uniform, which improves the efficiency of the subsequent network model learning dataset and improves the robustness of the algorithm to a certain extent.

### 3.3. Data Enhancement

In deep learning, the number of samples is generally required to be sufficient, and if the other conditions are the same, then the larger the number of samples used, the better the effect on the trained model. Since the amount of data used in this method is relatively small, data enhancement strategies are used. Through comparison and analysis, this chapter adopts two data augmentation strategies, horizontal flipping and vertical flipping, to improve the generalization performance and robustness of the model. The data enhancement effect diagram is shown in Figure 3.

### 3.4. Pretrained Model

In the field of dermoscopy image detection, most researchers currently use a single convolutional neural network for image classification, which has limited feature recognition ability, resulting in poor classification performance of the model. In this paper, three deep convolutional neural networks with different structures are selected as pre-training models to extract diverse feature information and improve the classification ability of dermoscopy images.

The first model chosen for our approach is the pretrained Xception model [30], a deep CNN architecture developed by Google researchers with a total of 71 layers. Figure 4 shows the structure of the Xception model. It is an improved version of the InceptionV3 model that mainly uses depthwise separable convolutions to replace the convolutions in Inception. The Xception model is formed by the linear superposition of depthwise separable convolutions. It can be divided into three parts: entry flow, middle flow, and exit flow, with a total of 14 modules. Except for the first and last modules, the remaining modules use residual connections.

The second model is a pretrained ResNet50 model [31]. The Microsoft research team developed ResNet in 2015. The network has several versions with different layers, and the common ones contain 18 layers, 50 layers, 101 layers, etc. Considering the small amount of training data and the low error-rate achieved by the ResNet50 on the ImageNet natural image library, the method in this paper selects the ResNet50 model. The ResNet50 has a total of 16 residual modules; each residual module contains 3 convolutional layers for a total of 49 convolutional layers, including 1 fully connected layer using the ReLU activation function and maximum and average pooling layers. The advantage of the ResNet50 is that it has a different residual module than the other networks. The residual part solves the problem of training a truly deep architecture by introducing skip-connections so that each layer can copy its input to the next layer. The building blocks of the ResNet50 model are shown in Figure 5. This jump-type structure can significantly alleviate the problem of stochastic gradient disappearance with the deepening of the network layer so that this network architecture can effectively reduce the overfitting phenomenon while extracting deeper features.

The third model is the pretrained Vgg16 model [32], which achieves a top-five error rate of 9.9% on the ImageNet natural image library. The structure diagram of the Vgg16 model is shown in Figure 6. The Vgg16 model has a total of 16 layers, including 13 convolutional layers and 3 fully connected layers. After 2 convolutions with 64 convolution kernels during the first time, pooling is used; after 2 convolutions with 128 convolution kernels during the second time, pooling is used again; and after 3 convolutions with 256 convolution kernels during the third time, one pooling is used again. Subsequently, 3 convolutions with 512 convolution kernels are repeated twice, followed again by pooling, and finally, three full connections. The entire network of the Vgg16 model is very regular, which can generate fewer parameters in the process of linear transformation, making the Vgg16 converge faster and thus effectively reducing the overfitting phenomenon.

### 3.5. Transfer Learning and Fine-Tuning Convolutional Neural Networks

Due to the small amount of data in medical image datasets, the generalization ability of the model is not good because it is very difficult to train a deep-learning model on a small amount of data, but this problem can be solved by a pretrained model. Transfer learning in convolutional neural networks is achieved by first training on a source domain of a large amount of data, commonly the ImageNet dataset, and then training and fine-tuning the weights of the convolutional neural network on a related but different target domain. Although there is no correlation between the natural image domain and the medical image domain, research shows that transfer learning can still work. By using the parameters of the pretrained model to initialize the weights and fine-tune the convolutional neural network, the problem of overfitting caused by less training data can be effectively solved.

In transfer learning and fine-tuning, the proposed method improves the structure and the fine-tuning strategy of the pretrained model. In the classification of the dermoscopic images, most of the transfer learning used is to change the number of output nodes of the fully connected layer of the pretrained model to the number of classification targets and then to retrain the fully connected layer of the pretrained model in order to achieve model migration. However, to train the final model, the pretrained model is based on the ImageNet dataset. The ImageNet dataset is comprised of natural images and is quite different from the dermoscopic images, with only the lower feature layers having good generalization performance, while the feature layers of the higher layers are quite different. However, the difference is large, resulting in poor transfer learning effect. Compared with the traditional algorithm, this paper first optimizes and improves the fully connected layer structure of the pretraining model because the fully connected layer of the pretraining model is built for the ImageNet dataset. In the classification problem, the number of fully connected layers and the number of neuron nodes in each layer are large, and a large number of parameters need to be calculated during the training process, when it is easy for overfitting to occur. However, in the two-classification problem of the dermoscopic images, there is no need for such a complex neural network as the fully connected layer of the model, so this paper re-optimizes and builds the fully connected layer of the three pre-trained models (Xception, ResNet50 and VGG16), as shown in Figure 7, which contains a 4-layer neural network with 8 neurons, and uses the dropout algorithm and L2 regularization to reduce the overfitting phenomenon of the model, effectively reducing the calculation parameters, speeding up the training speed, and improving the model’s classification performance.

The model is then trained by optimizing the fine-tuned policy. The first stage first freezes the convolutional layers of the pretrained model and only performs transfer learning training on the improved, fully connected layers. It is necessary to train the improved fully connected layers before fine-tuning so that the parameters of the fully connected layer of the model can quickly learn weights suitable for model fine-tuning before the fine-tuning, which improves the training speed and recognition ability of the model. In the second stage, the model is fine-tuned and trained to make it more data-specific in order to be able to learn the difference between the dermoscopic images and the ImageNet images and improve the classification performance of the model.

### 3.6. Integration of Models

In the field of dermoscopic image classification, most algorithms currently only use a certain convolutional neural network. Due to the limited ability of a single convolutional neural network to extract features, the classification performance is poor. The basic idea of ensemble learning is to first train several weak classifiers with certain differences, and then, integrate these weak classifiers through a certain ensemble strategy to form a strong classifier so as to achieve the effect of enhancing the generalization performance of the model [33]. Therefore, this paper proposes a dermoscopy image classification algorithm based on an integrated convolutional neural network. The structures and depth of the Xception, ResNet50 and VGG16 models are very suitable for medical image classification tasks with few training samples, and their overfitting phenomenon is relatively low. Additionally, the quality of the extracted features is better, the training speed is faster, and it has better classification performance. The structures of the Xception, ResNet50, and VGG16 models are also quite different but are very in line with the characteristics of the selected model. Therefore, they are selected as the basic models for the ensemble model in this paper. First, the three basic models of Xception, ResNet50 and VGG16 are trained on the dataset through the improved transfer learning and fine-tuning strategy in this paper, and then the ensemble model is obtained by adopting the integration strategy of weighted fusion. Finally, the model is compiled by setting the loss function, optimizer, and learning rate and evaluated on the test dataset.

### 3.7. Loss Function

The obvious class imbalance problem in the dataset used in this paper, i.e., that there are many benign melanoma samples, leads to slow model training and affects the prediction accuracy of the model. To solve this problem, the focal loss function for target detection is applied to the model proposed in this paper. The focal loss function is modified from the standard cross-entropy loss. This function can increase the model focus for the hard-to-classify samples during training by reducing the weight of the easy-to-classify samples. The formula for the cross entropy loss function is shown in Formula (1):(1)$$cross\_entropy\_loss=-y\log y^{\prime }-\left(1-y\right)\log\left(1-y^{\prime }\right)$$where y is the label and $y^{\prime }$ is the predicted value. 

The focal loss function adds two parameters to the cross-entropy loss function, as shown in Formula (2):(2)$$focal\_loss=-\alpha y{\left(1-y^{\prime }\right)}^{\gamma }\log y^{\prime }-\left(1-y\right)\log\left(1-y^{\prime }\right)$$
the parameter α is used to balance the influence of positive and negative samples on the loss value, and the parameter γ can make the model focus more attention on the indistinguishable samples and improve the final classification accuracy of the model.

### 3.8. Evaluation Metrics

As shown in Table 2, the confusion matrix is a commonly used method to evaluate and analyze the classification results of dermoscopic images. This paper uses four metrics, accuracy, precision, recall, and the F1-score to evaluate the classification performance of the model. Accuracy represents the accuracy of the prediction results, i.e., the number of correctly predicted samples divided by the total number of samples. Precision refers to the proportion of positive samples in the positive samples determined by the classifier. Recall represents the proportion of positive samples predicted as positive samples, and the F1-score represents the harmonic average evaluation index of precision and recall. The four indicators are defined as follows:(3)$$Accuracy=\frac{TP+TN}{TP+FN+FP+TN}$$
(4)$$Precision=\frac{TP}{TP+FP}$$
(5)$$Recall=\frac{TP}{TP+FN}$$
(6)$$F1-score=2*\frac{Precision*Recall}{Precision+Recall}$$

### 3.9. Experimental Platform and Parameter Settings

This paper uses the Google Colab experimental platform, based on the Python language environment, and uses the TensorFlow deep-learning framework to implement the research method of this paper. The specific model configuration is shown in Table 3. In this paper, the proposed method is trained and verified on the ISIC 2016 Challenge official skin dataset, and the Adam optimizer with a faster convergence speed and the Focal Loss function are used to optimize the model in this paper. In order to achieve the best performance of the model proposed in this paper, it was established through many experiments that the optimal number of training iterations of the model is set to 50, the batch size is set to 24, and the initial learning rate of the Adam optimizer is set to 1 × 10^−4^. This paper also uses ModelCheckpoint and ReducelRonplation from Keras to adjust the learning rate. ModelCheckpoint monitors performance indicators and regularly saves the model according to monitoring indicators such as validation loss. If the validation loss does not improve during model training, ReduceRonplation reduces the learning rate.

## 4. Results

### 4.1. Comparison of Effects before and after Integration

In this paper, by reconstructing the structure of the fully connected layers of the three pre-training models, Xception, ResNet50, and Vgg16, transfer learning and fine-tuning are performed with the ISIC 2016 Challenge official skin dataset, and then the outputs of these three basic models are weighted and fused to obtain an integrated model. The accuracy curves of the three basic models during training are shown in Figure 8.

As can be seen from Figure 8, in the first 20 epochs, the accuracy of the Xception model gradually improves, and the distance between the validation set curve and the training set curve gradually increases, which shows that the Xception model is in the process of learning the data features and is constantly improving. After the training process of 20 epochs, the accuracy curve of the Xception model begins to rise slowly and become more and more stable; at the same time, the distance between the training set curve and the validation set curve becomes smaller and smaller, and they finally coincide with each other, showing a good training effect. The accuracy of the ResNet50 model maintains an upward trend in the training process and finally becomes stable. The distance between the training set curve and the validation set curve is always small, and there is no obvious overfitting phenomenon, which shows that the model has strong generalization ability. In the first 25 epochs, the accuracy of the Vgg16 model shows an upward trend on the whole, and after the first 25 epochs, the accuracy undergoes changes in a small range and finally becomes stable. The distance between the validation set curve and the training set curve is widening, indicating that the generalization ability of the model needs to be improved in the training process. Loss curves for the training processes of the three basic models are shown in Figure 9.

As can be seen from Figure 9, the loss curves of the three basic models decrease with the increase in the epoch times and finally stabilize in a low numerical range, showing a good learning effect. The Xception model starts to converge after about 20 epochs of training, the ResNet50 model starts to converge after about 30 epochs of training, and the Vgg16 model starts to converge after about 30 epochs of training; there is no overfitting phenomenon in the three models. The confusion matrix of the three basic models on the test set is shown in Figure 10.

By fine-tuning and training three pre-trained models of Xception, ResNet50, and Vgg16 on the ISIC 2016 Challenge official dataset, the best basic model is obtained. Then, the output weighted fusion method of the three best base models is formed into an ensemble model, and finally, the model is compiled by setting the loss function of the ensemble model to focal loss, the optimizer to Adam, and the learning rate to 1 × 10^−4^, and the model is tested and evaluated on the test set of the ISIC 2016 Challenge dataset. Table 4 below shows the comparison of the accuracy rates under the different weighted combinations, where A represents the Xception model, B represents the ResNet50 model, and C represents the Vgg16 model. Table 5 is the comparison of the experimental results of the basic model and of the integrated model on the test set.

The results in Table 4 show that the accuracy of the integrated model using the {0.3A, 0.5B, 0.2C} combination is the highest, reaching 86.91%. In the integrated model, different fusion methods also have a certain impact on the recognition ability of the integrated model. In this paper, comparative experiments are carried out for three different fusion methods: output category voting, output category probability average, and output category probability weighting. The experimental results are shown in Table 5.

According to the experimental results in Table 5, the weighted fusion method for the model output probability performs best in accuracy, which shows that the weighted fusion method for the model output probability is a more reasonable fusion method, better combining the diversity of the model and better highlighting its advantages. The comparison of the experimental results between the basic model and integrated model is shown in Table 6.

Combined with the experimental results in Table 5 and Table 6, it is obvious that the accuracy of the ensemble model obtained by the three fusion methods is higher than that of the basic model, which further shows that the integrated model can integrate the prediction ability of the different models.

As can be seen from Table 6, the ensemble model of the integrated fine-tuned convolutional neural network significantly improves the accuracy, precision, recall, and F1-scores in the classification problem of dermoscopic images compared with the single model and significantly improves the classification effect of the dermoscopic images.

### 4.2. Performance Comparison of the Proposed Method with Other State-of-the-Art Methods

In order to further prove the advantages of the algorithm proposed in this study, the experimental results of the top five participating teams in the ISIC 2016 Challenge for the classification of melanoma images of the International Skin Imaging Collaboration Organization were selected to compare with the experimental results of the method in this paper; the results are shown in Table 7.

As can be seen from Table 7, the accuracy of the methods proposed in this paper are significantly improved and are higher than those of the other participating teams. This shows that the dermoscopy image classification method based on fine-tuning convolution neural networks proposed in this paper is effective and alleviates the impact of small and unbalanced data on the task of classifying the dermoscopic image to a certain extent. At the same time, the performance of the algorithm is compared with that of other algorithms in the literature, and the results are shown in Table 8.

In Table 8, this paper uses the same dataset as other comparative algorithms in the literature. It is not difficult to see that this method achieves the best results in the four common evaluation indexes and the results are significantly better than the methods in other studies. Among them, the accuracy index of this method is 0.61% higher than that of study [37]; In terms of the precision index, the method in this paper is 3.79% higher than that in the literature [34]; in the recall index, the method in this paper is 2.23% higher than that in study [36]; in terms of the F1-score index, the method in this paper is 2.25% higher than that in study [36]. This comprehensively shows that the method in this paper indicates a good prospect for the field of dermoscopy image classification and proves the effectiveness and feasibility of the method in this paper.

## *5.* Discussion

Of all the skin cancers, melanoma is the deadliest. Early detection and treatment are the most effective ways to cure melanoma, but the automatic classification of melanoma is a challenging task. In this paper, a dermoscopic image classification method integrating fine-tuned convolutional neural networks is proposed, and the feasibility and effectiveness of the method are verified through multiple experiments. The proposed method mainly reconstructs the fully connected layer of the pre-trained model, performs transfer learning and fine-tuning on the experimental dataset used in this paper, and then obtains the ensemble model through a weighted fusion strategy, finally evaluating the performance of the ensemble model on the test set. First of all, this paper determines the best weighted fusion method through many experiments so that the classification performance of the ensemble model can reach the best possible result; secondly, it is proved through experiments that the classification performance of the ensemble model is better than the basic model, and the indicators are significantly improved. The algorithm in this paper is compared with methods from the ISIC 2016 Melanoma Classification Challenge, and the results show that the performance of the proposed algorithm is better than the other methods and is helpful in improving the classification ability of dermoscopic images.

Although the algorithm proposed in this paper indicates good results, it also has shortcomings. The image preprocessing method of the algorithm in this paper is relatively simple and does not play a role in significantly improving the classification performance of the subsequent model. Recently, many studies [38,39,40] have discussed this issue. A good preprocessing method can effectively improve the training efficiency of the model and performance, which is also the future work direction of this research. In addition, due to the irregularity, similarity, and low contrast of the dermoscopic images, which hinder the classification of those dermoscopic images, follow-up work needs to further optimize the algorithm so that it can extract deeper features, and the dataset needs to be expanded. A greater number of images can provide the algorithm with more learning samples for training so as to improve its classification ability.

## 6. Conclusions

This paper proposes a dermoscopic image classification method based on an integrated fine-tuning convolutional neural network. Three pre-training models, Xception, ResNet50, and Vgg16 are selected as the basic models. By rebuilding the fully connected layer of the pre-training model, the official skin dataset of ISIC 2016 Challenge is used. Through the transfer learning and fine-tuning of the set and by adding a full fusion strategy, an integrated model was finally obtained. The experimental results show that the classification performance of the ensemble model is significantly improved compared to the base model, and it is better than some other traditional methods. It proves the feasibility of the proposed algorithm, which has a good effect on the classification of dermoscopic images and provides assistance, to a certain extent, in the diagnosis of melanoma. In future research work, we will continue to (i) optimize the model structure and classification algorithm to improve the performance of the model; (ii) focus on the image preprocessing algorithm to improve the effect of image preprocessing on the subsequent model classification; and (iii) expand the dataset to fully train the deep-learning model and improve the model generalization ability and classification performance.

## Figures and Tables

**Figure 1 sensors-22-04147-f001:**
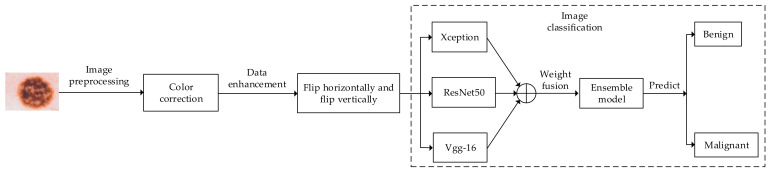
Flowchart of the proposed algorithm.

**Figure 2 sensors-22-04147-f002:**
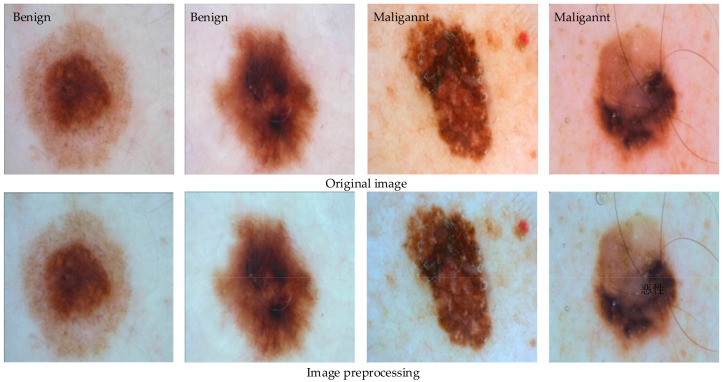
Image preprocessing.

**Figure 3 sensors-22-04147-f003:**
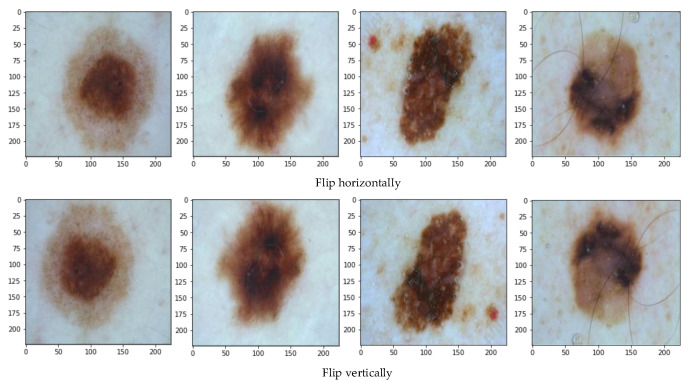
Data enhancement.

**Figure 4 sensors-22-04147-f004:**
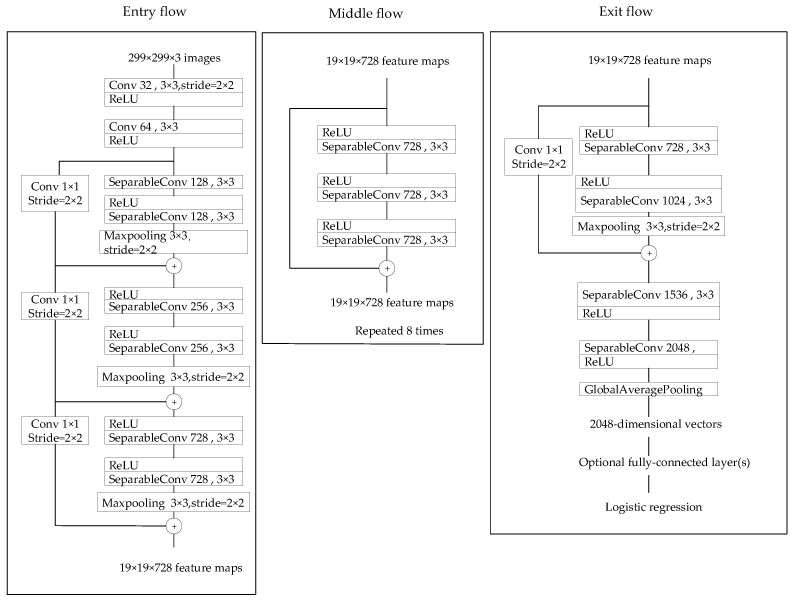
Structure diagram of Xception model.

**Figure 5 sensors-22-04147-f005:**
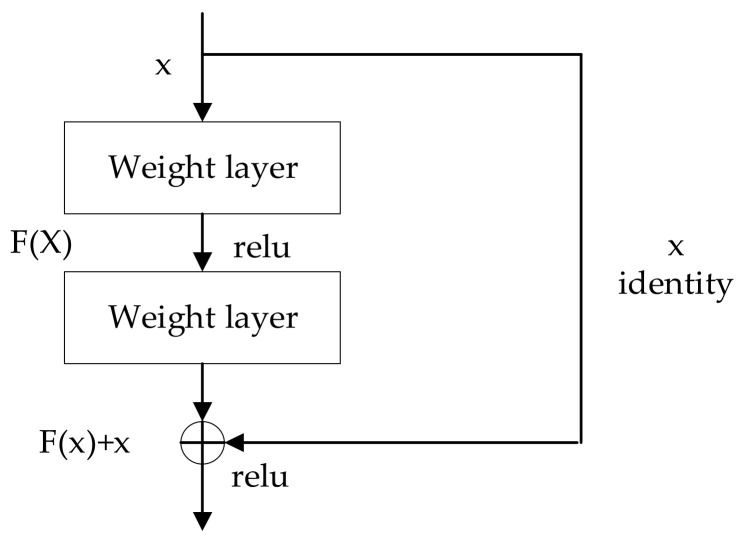
The building blocks of ResNet50.

**Figure 6 sensors-22-04147-f006:**
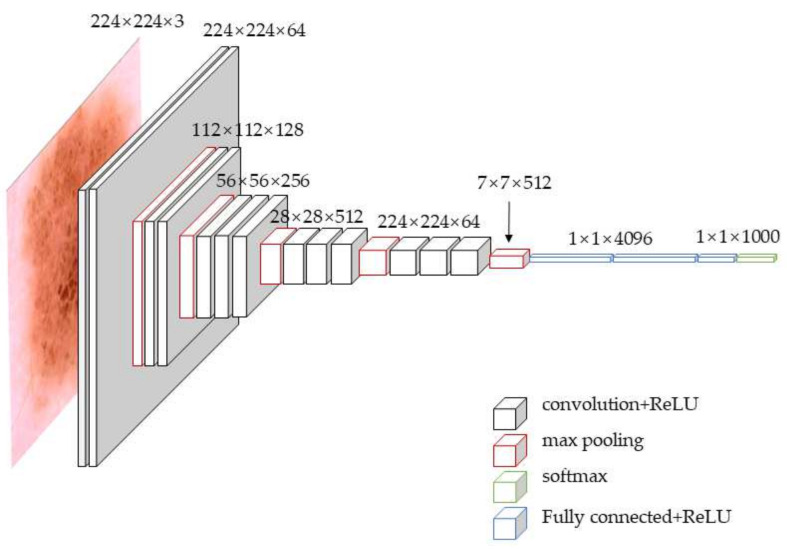
Structure diagram of Vgg16 model.

**Figure 7 sensors-22-04147-f007:**
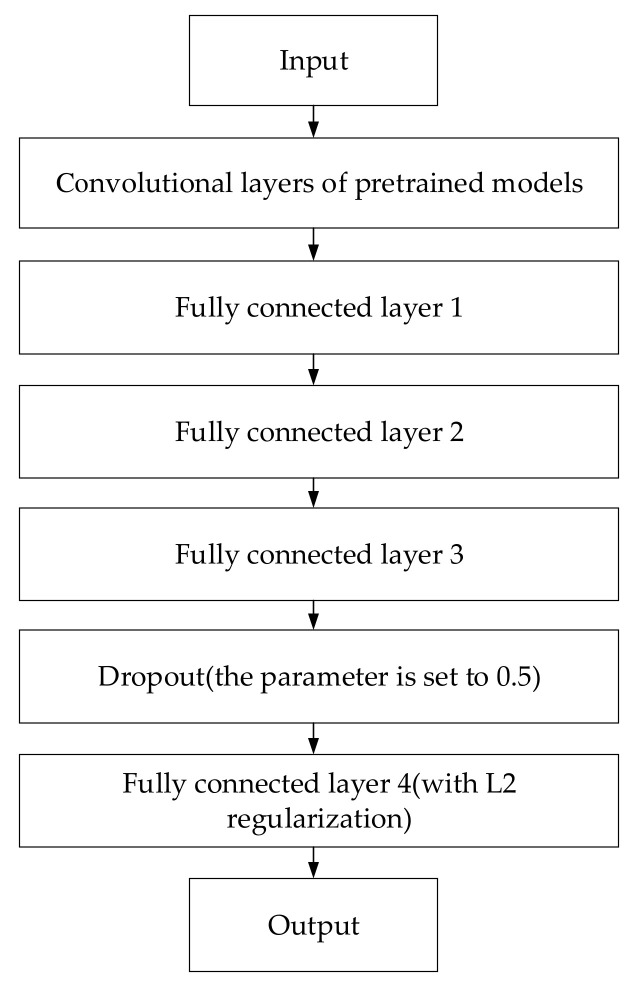
Improvement of model structure.

**Figure 8 sensors-22-04147-f008:**
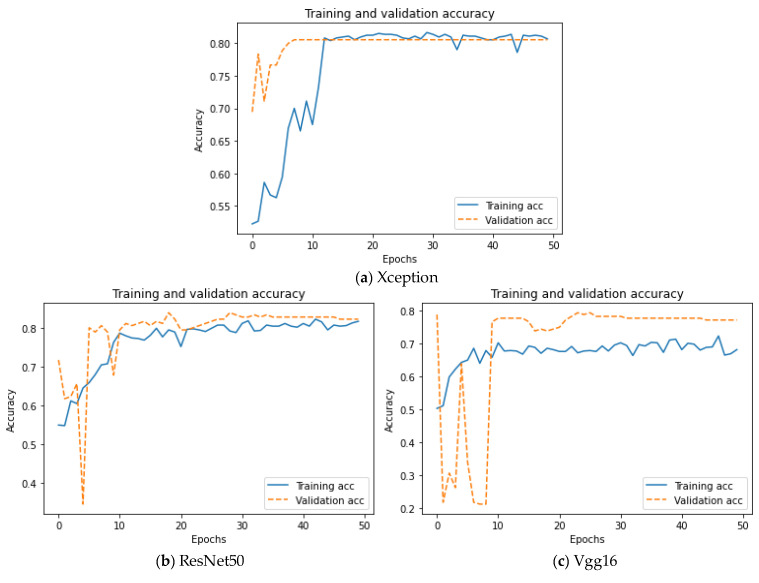
Accuracy curves of the three basic models.

**Figure 9 sensors-22-04147-f009:**
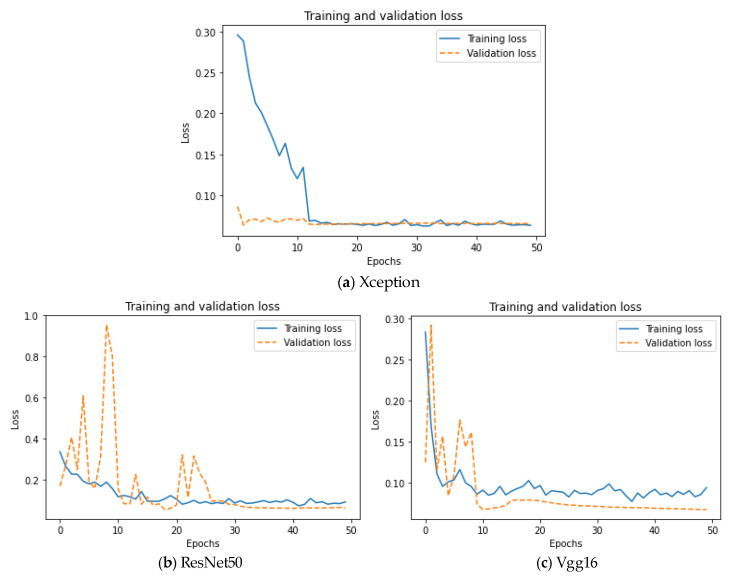
Loss curves of the three basic models.

**Figure 10 sensors-22-04147-f010:**
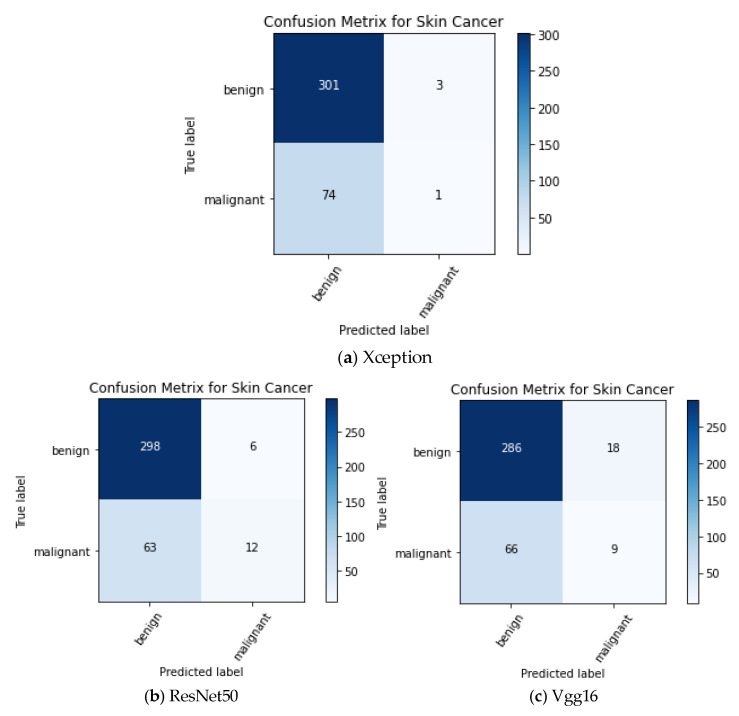
Confusion matrix for the three base models.

**Table 1 sensors-22-04147-t001:** Distribution of two types of dermoscopic images in the experimental dataset.

Dataset	Benign	Malignant	Total
Training	584	137	721
Validation	145	34	179
Test	304	75	379

**Table 2 sensors-22-04147-t002:** Confusion matrix.

Label	Prediction	
	True	False
Positive	TP	FP
Negative	FN	TN

**Table 3 sensors-22-04147-t003:** Parameter settings.

Parameters	Values
Experimental platform	Google Colab
Language	Python
Experiment framework	TensorFlow
Optimizer	Adam
Loss function	Focal Loss
Epochs	50
Learning rate (initial)	1 × 10^−4^
Batch size	24

**Table 4 sensors-22-04147-t004:** Accuracy of different model combinations.

Combination Mode	Accuracy
{0.4A, 0.3B, 0.3C}	84.78%
{0.3A, 0.4B, 0.3C}	85.30%
{0.3A, 0.5B, 0.2C}	**86.91%**

**Table 5 sensors-22-04147-t005:** Comparison of results of different fusion methods.

Fusion Method	Accuracy
Output category voting	84.57%
Output category probability average	85.65%
**Output category probability weighting**	**86.91%**

**Table 6 sensors-22-04147-t006:** Comparison of experimental results between the basic model and the ensemble model.

Model	Accuracy	Precision	Recall	F1-Score
Xception	80.56%	83.38%	82.15%	82.76%
ResNet50	83.89%	84.75%	81.86%	83.28%
Vgg16	79.44%	81.55%	80.11%	80.82%
**Ensemble Model**	**86.91%**	**85.67%**	**84.03%**	**84.84%**

**Table 7 sensors-22-04147-t007:** Comparison of ISIC 2016 Challenge competition results.

Method	Accuracy
GUMED	85.5%
GTDL	81.3%
BF_TB	83.4%
ThrunLab	78.6%
Jordan Yap	84.4%
**The Proposed Method**	**86.91%**

**Table 8 sensors-22-04147-t008:** Performance comparison of the proposed method and other studies’ algorithms.

Method	Model	Accuracy	Precision	Recall	F1-Score
Kaur R. [34]	LCNet	81.41%	81.88%	81.30%	81.05%
Zhang J. [35]	SDL	86.28%	68.10%	-	-
Al-Masni, M. A. [36]	Inception-ResNet-v2	81.79%	-	81.80%	82.59%
Tang P. [37]	GP-CNN-DTEL	86.30%	72.80%	32.00%	-
**The Proposed Method**	**The Ensemble Model**	**86.91%**	**85.67%**	**84.03%**	**84.84%**

## Data Availability

The ISIC 2016 Challenge official skin dataset is available at https://challenge.isic-archive.com/data/ (accessed on 10 January 2022).

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
