# Peer review of "Dermoscopic Image Classification Method Using an Ensemble of Fine-Tuned Convolutional Neural Networks"

_sensors, 2022, doi:10.3390/s22114147_

Round 1
Reviewer 1 Report
This paper describes a novel method for skin cancer diagnosis based upon a new image analyzing technique. The technology described here has demonstrated high accuracy and is showing good potential.
Author Response
Thank you for your approval, and I have carefully revised the paper again.
Please refer to the attachment

Reviewer 2 Report
This paper presents an ensemble classification model, consisting of three deep learning networks, for classifying dermoscopic images in benign or malignant. The experimental result outperforms state-of-the-art efforts for the used dataset (ISIC 2016) as far as accuracy is concerned.
However, it has a number of problems.
INTRODUCTION
1. A "Related Work" section is missing. So, I suggest that the authors split "Introduction" into "Introduction" and "Related Work" sections and enhance both. In exiting introduction, the authors refer to a number of quite old related efforts, which are using non-deep learning classifiers. I think, they should remove most of them and introduce other more recent efforts. The use of deep learning approaches is not that new, to justify use of a deep learning approach as an innovative element of the paper. Below are some recent efforts that should be included in the related work section:
1) Khan, M.Q.; Hussain, A.; Rehman, S.U.; Khan, U.; Maqsood, M.; Mehmood, K.; Khan, M.A. Classification of Melanoma and Nevus in Digital Images for Diagnosis of Skin Cancer. IEEE Access 2019, 7, 90132–90144.
2) Mahbod, A.; Schaefer, G.;Wang, C.; Dorffner, G.; Ecker, R.; Ellinger, I. Transfer learning using a multi-scale and multi-network ensemble for skin lesion classification. Comput. Methods Prog. Biomed. 2020, 193, 105475.
3) Zhang, J.; Xie, Y.; Xia, Y.; Shen, C. Attention Residual Learning for Skin Lesion Classification. IEEE Trans. Med. Imaging 2019, 38, 2092–2103.
4) Ameri, A. A Deep Learning Approach to Skin Cancer Detection in Dermoscopy Images. J. Biomed. Phys. Eng. 2020, 10, 801–806.
5) Almaraz-Damian, J.-A.; Ponomaryov, V.; Sadovnychiy, S.; Castillejos-Fernandez, H. Melanoma and Nevus Skin Lesion Classification Using Handcraft and Deep Learning Feature Fusion via Mutual Information Measures. Entropy 2020, 22, 484.
6) Hartanto, C.A.; Wibowo, A. Development of Mobile Skin Cancer Detection using Faster R-CNN and MobileNet V2 Model. In Proceedings of the 2020 7th International Conference on Information Technology, Computer, and Electrical Engineerin (ICITACEE), Semarang, Indonesia, 24–25 September 2020; pp. 58–63.
7) Srinivasu, P.N.; SivaSai, J.G.; Ijaz, M.F.; Bhoi, A.K.; Kim, W.; Kang, J.J. Classification of Skin Disease Using Deep Learning Neural Networks with MobileNet V2 and LSTM. Sensors 2021, 21, 2852.
8) Vatsala Anand, Sheifali Gupta, Deepika Koundal, Soumya Ranjan Nayak, Janmenjoy Nayak and S. Vimal. Multi-class Skin Disease Classification Using Transfer Learning Model. International Journal on Artificial Intelligence Tools Vol. 31, No. 2 (2022) 2250029 (19 pages). https://doi.org/10.1142/S0218213022500282.
9) Kousis, I.; Perikos, I.; Hatzilygeroudis, I.; Virvou, M. Deep Learning Methods for Accurate Skin Cancer Recognition and Mobile Application. Electronics 2022, 11, 1294. https://doi.org/10.3390/electronics11091294.
On page 3, in the 2nd paragraph, the authors mention: "the current algorithms often use a single convolutional neural network, and the classification performance of the model is low.". This is not totally true. From the above, refs 8 and 9 use a single convolutional neural network and outperform state-of-the-art results, even those using ensembles. So, please, change it.
Another problem is what is the innovation introduced in the paper. Pre-trained and fine-tuned networks as well as data augmentation and ensembles have been used by other works too. The authors should make clear what is the (new) contribution of their paper.
MATERIALS AND METHODS
The authors use the ISIC2016 dataset, which is an old dataset. There are newer versions of this dataset, like ISIC2017, ISIC2018, HAM10000. The authors should explain why the use that dataset.
In section 2.2. the first two sentences are too long, thus create syntax and comprehension problems. Please, check all long sentences in the text and split them in more than one.
On page 7, near the end: "Only the lower feature layers have good generalization performance, while the higher feature layers have good generalization performance" -->
"Only the lower feature layers have good generalization performance, while the
higher feature layers does not have good generalization performance"
On page 9, in the formulas (1) and (2) "y" is used, whereas in the text "γ" is used. Please, change appropriately.
RESULTS
In Table 8, besides the "Method" column, put a column "Model", where the deep learning model/architecture is presented. Also, you should include extra state-of-the-art works to compare, like refs 5, 6, 7, 8, 9 above, and explain why yours is comparable or better (e.g. dataset differences).
CONCLUSIONS
Adapt conclusions according to the changes you may make in the main text of the paper.
Author Response
Thanks for your valuable suggestion. I will reply to your comments as follows, and carefully revise:
1.Your suggestion:"A "Related Work" section is missing. So, I suggest that the authors split "Introduction" into "Introduction" and "Related Work" sections and enhance both."
My reply:I divided the "Introduction" into two parts, "Introduction" and "Related Work", and added a description of some recent efforts, and marked them in red in the text.
2.Your suggestion:”In section 2.2. the first two sentences are too long, thus create syntax and comprehension problems. Please, check all long sentences in the text and split them in more than one.”
My reply:I divided the first two long sentences of the paragraph into short sentences, and I have carefully checked the full text and highlighted in red in the text.
3.Your suggestion:"On page 7, near the end: "Only the lower feature layers have good generalization performance, while the higher feature layers have good generalization performance" -->"Only the lower feature layers have good generalization performance, while the higher feature layers does not have good generalization performance"
My reply:I've changed it over and double checked, and highlighted in red in the text.
4.Your suggestion:"On page 9, in the formulas (1) and (2) "y" is used, whereas in the text "γ" is used. Please, change appropriately."
My reply:I've changed it over and double checked, and highlighted in red in the text.
5.Your suggestion:"In Table 8, besides the "Method" column, put a column "Model", where the deep learning model/architecture is presented. Also, you should include extra state-of-the-art works to compare, like refs 5, 6, 7, 8, 9 above, and explain why yours is comparable or better (e.g. dataset differences)."
My reply:I have added the "Model" column to Table 8, and then elaborated the results of table 8 in detail, which are marked in red in the text.
Please refer to the attachment.

Reviewer 3 Report
The authors of the paper presented a method for skin cancer classification. This method uses three different convolutional deep networks and combines them into an ensemble.
Overall the paper is very good. However, I have a few minor comments and suggestions.
First of all, it is not described how the weights are selected. I guess that it was a grid search - perhaps using some simple heuristic algorithm would have produced even better results.
This sentence makes no sense (section 2.5): Only the lower feature layers have good generalization performance, while the higher feature layers have good generalization performance.
Dropout is not a layer (even though it is implemented that way in Keras). It is a mechanism to turn off selected neurons during learning.
Accuracy for unbalanced data is inappropriate. In my opinion, it is better to use Cohen's Kappa.
Author Response
Thanks for your valuable suggestion. I will reply to your comments as follows, and carefully revise:
1.Your suggestion:"This sentence makes no sense (section 2.5): Only the lower feature layers have good generalization performance, while the higher feature layers have good generalization performance."
My reply:I've changed it over and double checked, and highlighted in red in the text.
2.Your suggestion:"Dropout is not a layer (even though it is implemented that way in Keras). It is a mechanism to turn off selected neurons during learning."
My reply:I've changed it over and double checked, and highlighted in red in the text.
Please refer to the attachment.

Round 2
Reviewer 2 Report
The authors have satisfied most of my suggestions, except a few ones (see below).
INTRODUCTION
In the initial review report there was the following suggestion:
"On page 3, in the 2nd paragraph, the authors mention: "the current algorithms often use a single convolutional neural network, and the classification performance of the model is low.". This is not totally true. From the above, refs 8 and 9 use a single convolutional neural network and outperform state-of-the-art results, even those using ensembles. So, please, change it."
The authors have not addressed this comment. So, I rephrase it according to the new version of the paper:
On page 2, after line 12, the authors mention: "however, the current algorithms often use a single convolutional neural network, and the classification performance of the model is low". Please, rephrase it to: "however, the current algorithms often use a single convolutional neural network, and the classification performance of the model is low, with very few exceptions (like [39], [40])" (see below for [39], [40])
RELATED WORK
* They included in the related work only three of the nine (9) suggested new references. They left out the two most recent and with the best results on the HAM10000 dataset. Please, add those two references:
[39] Kousis, I.; Perikos, I.; Hatzilygeroudis, I.; Virvou, M. Deep Learning Methods for Accurate Skin Cancer Recognition and Mobile Application. Electronics 2022, 11, 1294. https://doi.org/10.3390/electronics11091294.
[40] Vatsala Anand, Sheifali Gupta, Deepika Koundal, Soumya Ranjan Nayak, Janmenjoy Nayak and S. Vimal. Multi-class Skin Disease Classification Using Transfer Learning Model. International Journal on Artificial Intelligence Tools Vol. 31, No. 2 (2022) 2250029 (19 pages). https://doi.org/10.1142/S0218213022500294.
MATERIALS AND METHODS
The authors haven't given an explanation about why they use the ISIC2016 set and not a newer version. They could say e.g. that a smaller dataset is more challenging (see for example in Ref [32], where a CNN is used for ISIC2016, 2017 and 2020, having worse results for ISIC2016 dataset).
Author Response
Thanks for your valuable suggestion. I will reply to your comments as follows, and carefully revise:
1.Your suggestion:On page 2, after line 12, the authors mention: "however, the current algorithms often use a single convolutional neural network, and the classification performance of the model is low". Please, rephrase it to: "however, the current algorithms often use a single convolutional neural network, and the classification performance of the model is low, with very few exceptions (like [39], [40])" (see below for [39], [40]).
My reply:I've changed it over and double checked, and highlighted in red in the text.
2.Your suggestion:They included in the related work only three of the nine (9) suggested new references. They left out the two most recent and with the best results on the HAM10000 dataset. Please, add those two references:
[39] Kousis, I.; Perikos, I.; Hatzilygeroudis, I.; Virvou, M. Deep Learning Methods for Accurate Skin Cancer Recognition and Mobile Application. Electronics 2022, 11, 1294. https://doi.org/10.3390/electronics11091294.
[40] Vatsala Anand, Sheifali Gupta, Deepika Koundal, Soumya Ranjan Nayak, Janmenjoy Nayak and S. Vimal. Multi-class Skin Disease Classification Using Transfer Learning Model. International Journal on Artificial Intelligence Tools Vol. 31, No. 2 (2022) 2250029 (19 pages). https://doi.org/10.1142/S0218213022500294.
My reply:I have added descriptions of these two references to the related work,and highlighted in red in the text.
3.Your suggestion:The authors haven't given an explanation about why they use the ISIC2016 set and not a newer version. They could say e.g. that a smaller dataset is more challenging (see for example in Ref [32], where a CNN is used for ISIC2016, 2017 and 2020, having worse results for ISIC2016 dataset).
My reply:According to your suggestion,I have explained the problem in the text,and highlighted in red in the text.
Please refer to the attachment.

This manuscript is a resubmission of an earlier submission. The following is a list of the peer review reports and author responses from that submission.